Foreground separation knowledge distillation for object detection

Li Chao chaoli@jiangnan.edu.cn 1
Liu Rugui 1
Quan Zhe 1
Hu Pengpeng 2
Sun Jun 1
1 School of Artificial Intelligence and Computer Science, Jiangnan University , Wuxi , Jiangsu Province , China
2 Centre for Computational Science and Mathematical Modelling, Coventry University , Coventry , United Kingdom
Yang Jiachen
Electronic publication date: 2024 Nov 13
Publication date: 2024
Volume: 10
Electronic Location ID: e2485
Received 2024 Aug 1; Accepted 2024 Oct 16
Copyright: ©2024 Li et al.
Copyright year: 2024
Copyright holder: Li et al.
License: This is an open access article distributed under the terms of the Creative Commons Attribution License, which permits unrestricted use, distribution, reproduction and adaptation in any medium and for any purpose provided that it is properly attributed. For attribution, the original author(s), title, publication source (PeerJ Computer Science) and either DOI or URL of the article must be cited.
License URL: https://creativecommons.org/licenses/by/4.0/

Keywords: Knowledge distillation, Object detection, Foreground separation, Channel feature

Funding: Natural Science Foundation of Jiangsu Province BK20221068 Fundamental Research Funds for the Central Universities JUSRP123031 National Natural Science Foundation of China 62272202 61672263 This work was supported by the Natural Science Foundation of Jiangsu Province (No. BK20221068), the Fundamental Research Funds for the Central Universities (No. JUSRP123031), and the National Natural Science Foundation of China (Nos. 62272202, 61672263). The funders had no role in study design, data collection and analysis, decision to publish, or preparation of the manuscript.

==============================
In recent years, deep learning models have become predominant methods for computer vision tasks, but the large computation and storage requirements of many models make them challenging to deploy on devices with limited resources. Knowledge distillation (KD) is a widely used approach for model compression. However, when applied in the object detection problems, the existing KD methods either directly applies the feature map or simply separate the foreground from the background by using a binary mask, aligning the attention between the teacher and the student models. Unfortunately, these methods either completely overlook or fail to thoroughly eliminate noise, resulting in unsatisfactory model accuracy for student models. To address this issue, we propose a foreground separation distillation (FSD) method in this paper. The FSD method enables student models to distinguish between foreground and background using Gaussian heatmaps, reducing irrelevant information in the learning process. Additionally, FSD also extracts the channel feature by converting the spatial feature maps into probabilistic forms to fully utilize the knowledge in each channel of a well-trained teacher. Experimental results demonstrate that the YOLOX detector enhanced with our distillation method achieved superior performance on both the fall detection and the VOC2007 datasets. For example, YOLOX with FSD achieved 73.1% mean average precision (mAP) on the Fall Detection dataset, which is 1.6% higher than the baseline. The code of FSD is accessible via https://doi.org/10.5281/zenodo.13829676.

Introduction

In recent years, deep learning models have gained significant attention and have been extensively used to solve complex problems in many scenarios, especially in computer vision (Guo, Xu & Ouyang, 2023). However, many deep learning vision models generally require a significant amount of computing and storage resources to run, making their deployment on mobile devices challenging (Guo, Xu & Ouyang, 2023). To address this issue, many model compression approaches, such as pruning (Wu et al., 2016; Han, Mao & Dally, 2015), quantization (Ullrich, Meeds & Welling, 2017; Chen & Zhao, 2018), lightweight model design (Han et al., 2020; Zhao et al., 2024), and knowledge distillation (KD) (Hinton, Vinyals & Dean, 2015; Li et al., 2023), have been proposed to reduce the demands on computational power and memory storage by removing redundant parts of the network.

Among these approaches, KD, initially introduced by Hinton, Vinyals & Dean (2015), exhibits higher interpretability, flexibility, and adaptability compared to other model compression techniques, thereby garnering more attention and broader adoption. Generally, KD aims to train a lightweight student model to mimic a larger teacher model, enabling the student model to achieve the teacher’s powerful performance without incurring additional costs (Yang et al., 2023a). With the advent of new intelligent edge devices and the expansion of large-scale data, KD techniques are evolving towards more efficient and intelligent directions.

Based on the type of knowledge extracted, KD can be broadly categorized into feature-, response-, and relation-based distillation (Romero et al., 2014). Among these, response-based KD necessitates the presence of a classification head in the teacher model, and relation-based KD requires explicit mathematical relationships between the knowledge representations of the student and teacher models, thereby limiting the general applicability of both approaches. In contrast, feature-based KD primarily operates on the internal features extracted by the models, offering greater versatile as it is not constrained by the architecture of the teacher model or the specific nature of the task (Yang et al., 2023a). Therefore, this paper focuses on feature-based KD and introduces a novel method within this paradigm.

Feature-based KD, such as attentional transfer (AT) (Zagoruyko & Komodakis, 2016) and factor transfer (FT) (Kim, Park & Kwak, 2018), focuses on mimicking the teacher model’s attention maps and feature map similarities. A more recent approach, channel-wise distillation (CWD) (Shu et al., 2021), normalizes the feature maps of each channel to derive a soft probability map. This method emphasizes the most salient regions within each channel by minimizing the Kullback–Leibler (KL) divergence between the probability maps of the teacher and student models. However, when these KD algorithms are applied to solve object detection tasks, their effectiveness is frequently hindered by interference from background noise.

In order to obtain an effective detector, researchers have proposed various KD methods by considering the imbalance of information between foreground and background features. For example, Wang et al. (2019) considered the discrepancy of feature response on the near object anchor locations, enabling the student model to more precisely learn the positional information of the teacher model. The Focal and Global Distillation (FGD) (Yang et al., 2022b) utilizes a binary mask for foreground-background segmentation, forcing the student to focus on the teacher’s critical pixels and channels, and can be applied to various detectors. Yang et al. (2022a) investigated the impact of foreground and background features on KD, aiming to improve distillation performance by balancing the feature differences between foreground and background. The Masked Generative Distillation (MGD) method (Yang et al., 2022c) transforms the imitation task into a generative one. By randomly masking parts of the student’s features during distillation, MGD enables the student to reconstruct the entirety of the teacher’s features from merely a subset of its own, thereby improving the student’s representational capacity.

The characteristics and limitations of the KD methods for object detection, including the aforementioned feature-based approaches and the non-feature-based ones such as Cross-head KD (CrossKD) (Wang et al., 2024a) and Bridging Cross-task KD (BCKD) (Yang et al., 2023b), are categorized and listed in Table 1. Compared to feature-based KD methods that solely rely on feature maps, approaches that incorporate foreground-background separation via masks typically achieve superior performance in object detection tasks. Among these mask-based methods, FGD stands out by directly utilizing the ground-truth bounding box to define the foreground, resulting in more precise foreground feature extraction than other techniques (Yang et al., 2022b). However, these mask-based KD methods, including FGD, employ binary masks, compelling the student model to learn the bounding box information designated by the mask with high accuracy, which can significantly burden student models, particularly those with limited parameters. Additionally, all feature-based KD methods outlined in Table 1 only rely on spatial information. If the information from the features extracted by the network could also be integrated into the loss function, the accuracy of the student model could be further enhanced.

Table 1 Summary of KD methods for object detection tasks.

KD method type	Representative study	Characteristics	Limitations	
Non-feature-based KD	CrossKD (Wang et al., 2024a) BCKD (Yang et al., 2023b)	The student model learns from the teacher model’s output (response-based KD), or is trained based on the relational knowledge existing between the student and teacher (relation-based KD).	Response-based KD necessitates that the teacher model to have a classification head, while relation-based KD requires explicit mathematical relationships between the students and teachers.	
Feature-based KD using only feature maps	AT (Zagoruyko & Komodakis, 2016) FT (Kim, Park & Kwak, 2018) CWD (Shu et al., 2021)	The distillation loss is specifically formulated based on the feature maps extracted from both the teacher and the student.	These methods are prone to background noise in object detection tasks, which can negatively impact detection accuracy.	
Feature-based KD using both feature maps and masks	fine-grained feature imitation (Wang et al., 2019) ARSD (Yang et al., 2022a) MGD (Yang et al., 2022c) FGD (Yang et al., 2022b)	In addition to extracting feature maps, masks are applied to filter out finer-grained information for distillation.	Most of these methods utilize binary masks, which require student models to precisely learn the bounding box information specified by the mask. This demand can be overly demanding for student models with limited parameters.	
Notes.

CrossKD cross-head knowledge distillation

BCKD bridging cross-task knowledge distillation

AT attention transfer

FT factor transfer

CWD channel-wise distillation

ARSD adaptive reinforcement supervision distillation

MGD masked generative distillation

FGD focal and global knowledge distillation

Motivated by these challenges, and building on the state-of-the-art FGD method, this paper proposes a novel foreground separation distillation (FSD) method. The main contributions of this study are as follows:

(1) Like FGD, the FSD method separates foreground and background using the bonding box, but it enhances this process by applying a Gaussian heatmap to weight foreground features. This method alleviates the rigidity imposed by binary masks by providing more heatmap space for foreground features, enabling the student model to focus more effectively on foreground information and easing the learning of bounding box localization. As a result, the success rate of the student model in object detection tasks is significantly improved.

(2) The FSD method also integrates a spatial feature transformation module that transposes spatial foreground and background features into the channel dimension. Given that channel-level information often represents more salient features than spatial information, incorporating the KL divergence to the loss function to align the foreground and background features of the student and teacher models allows the lightweight student model to focus more on essential feature information.

(3) The proposed FSD method is applied to the YOLOX detector (Ge et al., 2021). Experimental results on both a Fall Detection dataset and the VOC2007 dataset demonstrate that FSD not only outperforms the several canonical and state-of-the-art KD algorithms, but also achieves significantly better performance than FGD in object detection tasks.

The remainder of this paper is organized as follows. ‘Related work’ introduces the related work pertinent to the studies presented in this paper. The foreground separation knowledge distillation is presented in ‘Foreground Separation Distillation’, followed by the experimental results and the corresponding discussions in ‘Experiments and Analysis’. Some conclusions and directions for future work are given in ‘Conclusions’.

Related Work

Focal and global knowledge distillation

The FGD method synergizes the strengths of focal and global KD to enhance the performance and generalization capability of the student model. Focal KD focuses on fine-grained feature transfer, enabling the student model to more efficiently learn and replicate the teacher model’s performance in specific areas or tasks. For example, the FitNets method (Romero et al., 2014) achieves effective distillation of deep networks by aligning intermediate layer features. This method allows the student model to learn the teacher model’s intermediate layer features, thereby better capturing detailed features, especially in image classification tasks. In contrast, global KD emphasizes the extraction and transfer of holistic features, focusing on the overall optimization of the model’s performance. The soft targets distillation method proposed by Hinton, Vinyals & Dean (2015) is a typical example of global KD, which employs softened probability distributions (soft targets) to guide the training of the student model, enabling the student model to comprehensively learn the decision boundaries of the teacher model. Another example is a self-distillation framework proposed by Wang et al. (2024b). This framework introduces additional logit output branches for each shallow block to ensure consistent feature map dimensions between the shallow classifiers and the deep classifier, mitigating potential information loss and its adverse effects.

FGD amalgamates these approaches, significantly improving the performance of the student model through multi-level knowledge transfer. Focal distillation in FGD separates foreground and background, forcing students to focus their attention on the teacher’s key pixels and channels. Global distillation in FGD reconstructs the relationships between different pixels and passes them from the teacher to the students to compensate for the lost global information in local distillation. This combined approach not only enhances the accuracy of the student model but also improves its adaptability to different tasks and data distributions.

YOLOX

YOLOX (Ge et al., 2021) significantly advances object detection by building on the YOLO (You Only Look Once) series’ foundations and incorporating key innovations. Unlike its predecessors, YOLOX adopts an anchor-free mechanism, which simplifies the detection pipeline, reduces computational overhead, and improves performance, especially for small and irregularly shaped objects. Additionally, YOLOX employs a decoupled head architecture, separating classification and localization tasks into distinct branches, facilitating independent optimization and better precision. Empirical results illustrate that YOLOX outperforms most previous YOLO models and other contemporary detectors in both speed and accuracy, making it highly suitable for real-time applications such as autonomous driving and video surveillance.

Given the aforementioned advantages of YOLOX, applying our proposed KD method to YOLOX would serve as an exemplary case study. Moreover, the baseline method used in this paper, FGD, has also been implemented on YOLOX (Yang et al., 2022b). Therefore, although FSD is applicable to various object detection models, this paper focuses on its implementation with the YOLOX detector to effectively demonstrate the superiority of FSD.

Foreground Separation Distillation

Overall framework of the foreground separation distillation method

The design of the KD algorithm critically influences the knowledge transformation capability of the teacher model, which is directly related to the final performance of the student model. To enhance the knowledge transfer capability of the model, the proposed FSD method first manipulates the feature space of the model to enhance the learning and comprehension abilities of the student. Subsequently, an attention mechanism is introduced to extract essential task-related information from attention maps of the teacher and transfer it to the student. This process ensures that the student network focuses more on foreground information and minimizes the impact of background noise on the model.

The overall framework of the FSD method is depicted in Fig. 1. The input image is fed into both the teacher network and the student network for feature extraction, and after which the feature maps are generated. The network differentiates between the foreground and background based on their feature differences. Then, the spatial features of the foreground and background from both the teacher and student are converted into corresponding channel features. Finally, these features are transformed into probabilistic forms, and the KL divergence is utilized to measure the differences between the feature maps.

Figure 1 FSD framework.

The raw image was obtained from the UR Fall Detection dataset. This dataset is licensed under a Creative Commons Attribution-NonCommercial-ShareAlike 4.0 International License and is intended for non-commercial academic use (http://fenix.ur.edu.pl/m˜kepski/ds/uf.html).

Foreground separation based on Gaussian mask

Considering the significant difference in model complexity between the teacher and the student, the FSD method incorporates a foreground feature extraction module to separate and individually distill the foreground and background. This module interprets and adapts the knowledge of the teacher model before transferring it to the student, thereby streamlining the training process and reducing the learning difficulty for the student model.

Most currently object detectors employ a Feature Pyramid Network (FPN) to fuse features at different scales. Since FPN can effectively fuse feature information from different levels of the backbone during KD on a detection model, the teacher network aggregates multi-level features through the FPN and subsequently transfer the learned feature knowledge to the student model, enhancing the student model’s performance. Generally, the feature distillation can be formulated as follows: (1) Lfea=1CHW∑k=1C ∑i=1H ∑j=1WfFk,i,jT,Fk,i,jS

where FT represents the output features of the teacher model, and FS denotes the output features of the student model. f(⋅) is a loss function to measure the discrepancy between the outputs of the student and teacher models. H and W represent the height and width of the feature maps, respectively. C denotes the number of channels in the feature maps.

Specifically, the FGD method uses a binary mask M to separate the foreground and background, as illustrated in Eq. (2): (2) Mi,j=1,ifi,j∈d0,otherwise

where d denotes the bounding box region, and i and j are the horizontal and vertical coordinates of the feature map. If the point (i, j) falls within the bounding box, then Mi,j = 1, otherwise it is 0.

In order to highlight the features of the foreground more prominently, different from FGD, we propose a novel foreground separation approach based on a Gaussian mask in this paper to separate foreground and background from the feature map obtained by the FPN, which can be expressed as follows: (3) Mi,j=1 ,ifi,j∈de−12x−x¯x¯2−y−y¯y¯2,ifi,j∈d ˆ0 ,otherwise

where a threshold region d ˆ is introduced, which represents a region that is twice the size of the bounding box around d. If the point (i, j) is within the threshold region d ˆ, then Mi,j satisfies Gaussian distribution. In other words, the area around the bounding box is diffused by using the Gaussian heatmaps according to Eq. (3).

Furthermore, to better separate the foreground and background, it is necessary to extract features from both. Therefore, we apply a binary mask to the background, which is defined as follows: (4) Mbg=1,ifi,j⁄∈d∪d ˆ0,otherwise

In different regions, pixels need to have different scores; pixels within the bounding box should have the highest scores, and the farther pixels are from the bounding box, the lower their scores. To achieve this, we design a mask S with varying scales: (5) Si,j=1HdWd,ifi,j∈d1HdWd,ifi,j∈d ˆ1Nbg,otherwise

According to Eq. (5), the proportional masks within the range of d and d ˆ are both set to 1HdWd, while the mask for the background region is set to 1Nbg.

In order to have a more intuitive experience of our proposed foreground separation approach, we illustrate the mask images of the same input by FGD and our approach, as shown in Figs. 2 and 3, respectively. Moreover, Fig. 4 demonstrates the heatmaps obtained by FGD and FSD. From Fig. 4, it can be seen that after applying the Gaussian heatmap diffusion, the heatmaps highlight the foreground features more prominently, facilitating easier learning for the student model.

Figure 2 Foreground and background mask images obtained by the binary mask in FGD.

(A) Input image. (B) Foreground mask image. (C) Background mask image. Raw input image source: UR Fall Detection dataset.

Figure 3 Foreground and background mask images obtained by the proposed Gaussian mask in FSD.

(A) Input image. (B) Foreground mask image. (C) Background mask image. Raw input image source: UR Fall Detection dataset.

Figure 4 Comparison of heatmaps after foreground separation obtained by (A) FGD and (B) FSD.

Channel feature extraction with probability distribution

Generally, KD methods calculate the total loss 𝔏KD by aggregating the loss of each pixel in the spatial dimension: (6) LKD=ly,yS+α⋅lϕyT,ϕyS

where y represents the ground truth, with yT and yS denoting the activation maps from the teacher and the student models, respectively. l(⋅) represents the loss function. ϕ(⋅) is a function that performs a transformation on features. Thus, for Eq. (6), the first term in the right part is the loss used to measure the difference between the output of the student model and the ground truth, and the second term in the right part represents the distillation loss between the student model and the teacher model. The factor α serves as an equilibrium factor between these two terms.

Activations in different channels encode various characteristics of different categories within the input image. Additionally, well-trained teacher networks for object detection can typically display activation maps of specific masks with clearly defined categories on each channel. Therefore, in this paper, we propose a channel feature extraction approach based on foreground and background separation. This approach transforms the corresponding spatial feature maps into probabilistic forms, in order to fully utilize the knowledge in each channel of a well-trained teacher. Specifically, we convert the activation of the spatial feature maps into a probability distribution, so that we can use probability distance metrics (such as KL divergence) to measure the differences in feature maps. To this end, the distillation loss on the channels can be specifically expressed as follows: (7) lϕyT,ϕyS=lϕycT,ϕycS

where c = 1, 2, …, C represents different channels, with yc being the feature map in channel form. If the features in a certain channel are determined to be foreground features, ϕ(⋅) should convert the activation values of the features in that channel into a probability distribution: (8) Φyc=expyc,iτ ∑i=1W⋅H expyc,iτ

where i denotes different spatial positions within each channel, and τ is a hyperparameter indicating the temperature. A higher τ leads to softer probabilities, resulting in a smoother probability distribution. This means that each channel of the feature map can attend to a wider spatial area and provide more contextual information, thereby effectively enhancing detection performance. Moreover, to mitigate the impact of scale differences between large and small network models, we apply softmax normalization to the feature maps. Softmax normalization is advantageous for KD since it makes the loss function of the model easier to optimize, thereby accelerating the convergence rate of the FSD. Additionally, this paper utilizes the KL divergence to evaluate the differences between the student and teacher networks according to the following equation: (9) lyT,yS=τ2C∑c=1C ∑i=1WHϕyc,iT logϕyc,iTϕyc,iS

Attention mechanism

Incorporating an attention mechanism during model training can enhance the model’s focus on the information critical to the current task. The effectiveness of this strategy is demonstrated by the SENet (Hu, Shen & Sun, 2018) and CBAM (Woo et al., 2018), which shows that assigning higher weights to some important channels can significantly improve model performance. In this paper, we adopt an attention mechanism similar to that in the study by Zagoruyko & Komodakis (2016) to select critical regions and channels. Specifically, we first calculated the absolute mean values for different pixels and channels: (10) GSF=1C∑c=1C|Fc|

(11) GCF=1H⋅W∑i=1H ∑j=1W|Fi,j|

where GS and GC are the spatial and channel attention maps on the feature map. Then, the attention mask can be calculated as follows: (12) ASF=H⋅W⋅softmaxGSFτ′

(13) ACF=C⋅softmaxGCF/τ′

where τ′ is the temperature hyperparameter introduced by Hinton et al. to adjust the distribution, with AS and AC representing the spatial attention mask and channel attention mask of the teacher detector, respectively.

Since there are significant differences between the masks of the student and the teacher models, we utilized the mask of the teacher model to guide the training process of the student. Specifically, the loss for foreground and background of the student LStufg and Lstubg, and those for the teacher Lteafg and Lteabg can be calculated by Eqs. (14) to (17), respectively. (14) Lstufg= ∑k=1C ∑i=1H ∑j=1WMi,jSi,jAi,jSAkCFk,i,jS

(15) Lstubg= ∑k=1C ∑i=1H ∑j=1W1−Mi,jSi,jAi,jSAkCFk,i,jS

(16) Lteafg= ∑k=1C ∑i=1H ∑j=1WMi,jSi,jAi,jSAkCFk,i,jT

(17) Lteabg= ∑k=1C ∑i=1H ∑j=1W1−Mi,jSi,jAi,jSAkCFk,i,jT

Subsequently, the feature loss Lfea can be calculated as follows: (18) Lfea=lLstufg,Lteafg+lLstubg,Lteabg

Finally, the total loss of the entire framework is (19) L=Lfea+Lat+Lglobal

where Lat is the attention loss and Lglobal is the global loss of the FSD method.

Experiments and Analysis

Dataset

In this study, the Fall Detection dataset and the Visual Object Classes (VOC) 2007 dataset were employed in our experiments. The dataset is available at the following link: https://zenodo.org/doi/10.5281/zenodo.12752189. Figures 5 and 6 display some examples from the Fall Detection dataset and VOC2007 dataset, respectively.

Figure 5 Examples for the Fall Detection dataset.

Raw input image source: UR Fall Detection dataset.

Figure 6 Examples for the VOC2007 dataset.

The Fall Detection dataset comprises the UR Fall Detection Dataset (Kwolek & Kepski, 2014) and the indoor Fall Detection dataset (Adhikari, Bouchachia & Nait-Charif, 2017). This dataset is formatted in COCO style and includes two categories: standing and falling. The dataset contains over 4,000 images, with 80% of the entire dataset randomly selected as the training set and 20% as the test set.

The PASCAL VOC 2007, originally a project initiated by the European Conference on Computer Vision (ECCV) and reported by Everingham et al. (2010), mainly used for object detection, image classification, and semantic segmentation tasks. In this study, we leverage the dataset to evaluate the object detection performance of the compared algorithms. The PASCAL VOC 2007 dataset contains a total of 9,963 images, which are re-divided into a 7:3 ratio to distinguish between the training set and the test set.

Experimental environment and parameter settings

All the experiments presented in this paper were conducted using the mmdetection framework (Chen et al., 2019), implemented in PyTorch running on a system with a 22-core AMD EPYC 7T83 CPU, NVIDIA GeForce RTX 4090 GPU and 90 Giga-bytes memory. The operating system is Ubuntu 20.04, with CUDA version 11.3 and Python version 3.8.

The parameter settings for all the experiments are as follows: the input image size is 640 × 640 pixels, the training epochs for the Fall Detection dataset are 120 while the training epochs for VOC2007 dataset are 40. Since all compared KD methods were applied to the YOLOX detector in this study, all the other parameters align with the default parameters used for YOLOX training, with an initial learning rate of 0.1, momentum set to 0.9, weight decay to 0.0005, and a batch size of 16 images per GPU. YOLOX-L was chosen as the teacher model and YOLOX-S served as the student model.

Evaluation metrics

In this paper, for both two dataset formats (COCO and VOC), the mean Average Precision (mAP) is adopted as the evaluation metric, where Average Precision (AP) represents the area under the Precision-Recall curve for a specific class. The formulas for calculating AP and mAP are shown as follows: (20) AP= ∫01Prdr

(21) mAP=∑APN

where P is the average precision value for the current class, and N is the number of sample categories in the dataset.

For the COCO-format Fall Detection dataset, we also report the mean Average Recall (mAR) and other AP-related metrics supported by the COCO format, including AP50, AP75, APS, APM, and APL. mAR is utilized to measure the whether the model can detect all positive samples and to identify any missed detections. AP50 and AP75 are special cases of calculating AP where the Intersection over Union (IoU) thresholds are set to 0.5 and 0.75, respectively. IoU is the ratio of the intersection over the union between the predicted bounding box and the ground truth box. A prediction is considered correct when the IoU is greater than or equal to a certain threshold (e.g., 0.5 or 0.75). The last three metrics measure the performance of detecting objects of different scales: small, medium, and large.

In addition, a new metric tailored for object detection, the optimal Localization Recall Precision (oLRP) error (Oksuz et al., 2018), has been introduced as an evaluation metric for the COCO-format Fall Detection dataset. The Localization Recall Precision (LRP) error consists of three components: localization, false negative (FN) rate and false positive (FP) rate, as detailed in Eq. (22). Compared to AP, LRP provides more detailed and discriminative information. (22) LRPY,Zs:=1NTP+NFP+NFN∑i=1NTP1−IoUyi,zyi1−τ+NFP+NFN

where Y is the set of the ground truth boxes, Z denotes the set of the predicted boxes returned by an object detector, and Zs represents the set of detections with confidence score larger than s.NTP, NFP and NFN are the number of true positives (TPs), FPs and FNs, respectively. LPR penalizes each TP for its erroneous localization normalized by 1 − τ to the [0,1] interval, each FP and FN by 1 (the maximum penalty). The sum of these errors is normalized by NTP + NFP + NFN, yielding a value that quantifies the average error per bounding box, ranging from 0 to 1. The lower the LRP error, the better the performance of the model.

Based on LRP, the oLRP is defined as the minimum achievable LRP error when τ = 0.5, representing the best balance between precision, recall, and IoU. Formally, oLRP can be expressed as: (23) oLRP:= minsLRPX,Ys.

Comparison of FSD and some state-of-the-art KD methods

In this section, the proposed FSD method is compared against several canonical and state-of-the-art KD methods on the Fall Detection dataset and VOC2007 dataset. The methods compared, as selected from Table 1, comprise three feature-based KD methods that solely rely on feature maps, two feature-based KD methods that utilize both feature maps and masks, and a state-of-the-art non-feature-based KD method. The results of comparisons are detailed in Tables 2 and 3.

Table 2 Comparison of different KD methods on the Fall Detection dataset.

KD method	mAP	mAR	AP50	AP75	APS	APM	APL	oLRP	
FGD	0.715	0.742	0.980	0.826	0.4	0.663	0.748	0.331	
AT	0.662	0.704	0.977	0.788	0.396	0.633	0.712	0.368	
FT	0.688	0.728	0.978	0.786	0.394	0.643	0.719	0.36	
MGD	0.713	0.750	0.980	0.829	0.455	0.672	0.741	0.334	
CWD	0.716	0.764	0.957	0.799	0.328	0.663	0.753	0.373	
BCKD	0.714	0.762	0.959	0.812	0.366	0.655	0.752	0.375	
Ours	0.731	0.768	0.981	0.832	0.431	0.675	0.763	0.311	
Notes.

FGD focal and global knowledge distillation

AT attention transfer

FT factor transfer

MGD masked generative distillation

CWD channel-wise distillation

BCKD bridging cross-task knowledge distillation

Table 3 Comparison of different KD methods on the VOC2007 dataset.

KD method	mAP	
FGD	0.3365	
AT	0.2937	
FT	0.2753	
MGD	0.4248	
CWD	0.3872	
BCKD	0.3813	
Ours	0.3904	
Notes.

FGD focal and global knowledge distillation

AT attention transfer

FT factor transfer

MGD masked generative distillation

CWD channel-wise distillation

BCKD bridging cross-task knowledge distillation

The experimental results in Table 2 indicate that the proposed FSD method achieved the highest performance in both mAP and mAR, demonstrating superior overall precision and recall compared to other methods. Specifically, for AP50 and AP75, our algorithm outperformed other KD methods, with a noticeable advantage in AP75, indicating that FSD remains highly effective when more precise object localization is required. This suggests that the proposed foreground separation approach using the Gaussian mask facilitated more efficient learning of foreground features by the student model compared to the approaches using binary masks. In terms of detecting objects of different sizes, FSD achieved the best results for medium and large objects, particularly excelling in large-object detection. This success is likely attributed to the spatial-to-channel transformation module, which effectively filtered out irrelevant features that could impair object detection, enabling the student model to focus more on critical foreground features. For small-object detection, FSD also performed well, ranking second only to MGD. Moreover, in terms of the comprehensive evaluation metric, oLRP, FSD achieved the lowest oLRP score of 0.311 among all methods, highlighting its superior balance between localization accuracy, recall, and precision. Overall, our method outperformed all the other compared KD methods across almost all performance metrics, especially demonstrating strong robustness across different IoU thresholds and object sizes. Additionally, FSD significantly outperformed the baseline FGD across all metrics, further proving the effectiveness of the proposed modules in this paper.

The results on the VOC2007 dataset (Table 3) also demonstrate that the FSD method is better than most of the compared methods, with a mAP improvement of nearly 6% than the baseline FGD. The exception is the MGD method. Unlike the other mask-based KD methods shown in Table 1, MGD applies masks to the student features, compelling the student to generate more robust teacher-like features using only a subset of its own features. However, the comparative and matching requirements of the hidden layer features between the teacher and student models may necessitate additional computational resources and extended training times. In contrast, our proposed method requires fewer computational resources yet still achieves performance similar to that of the MGD.

Moreover, to provide a more intuitive understanding of the performance differences among the compared algorithms, we plotted the changes of mAP in terms of the number of iterations obtained by each comparative algorithm on both two datasets, as illustrated in Fig. 7. From Fig. 7A, it can be seen that our proposed method has a noticeable increase in mAP at the beginning of the training, followed by a slower growth rate. Around 20,000 iterations, our method surpasses the other comparative algorithms and maintains the lead thereafter. For the results on the VOC2007 dataset, although our method does not surpass MGD, it demonstrates a more stable training process. Specifically, the MGD method experiences a noticeable decline in mAP after reaching its peak, while the FSD maintains a more consistent performance.

Figure 7 Changes of mAP in terms of the number of iterations obtained by each compared method on the (A) Fall Detection dataset and (B) VOC2007 dataset.

Ablation experiments

To further validate and analyze the effectiveness and robustness of each module in the FSD method, several ablation experiments were conducted based on the Fall Detection dataset. Given that our work builds upon the FGD method, we adopted the FGD as the baseline. The tested modules include the Gaussian mask foreground separation (GMFS) approach and the channel feature extraction (CFE) approach, as discussed in ‘Foreground separation based on Gaussian mask’ and ‘Channel feature extraction with probability distribution’, respectively. The results on the Fall Detection dataset and the VOC2007 dataset are recorded in Tables 4 and 5, respectively. Moreover, similar to ‘Comparison of FSD and some state-of-the-art KD methods’, we also plotted in Fig. 8 the changes of mAP during the training process by each compared method.

Table 4 Ablation study on the effectiveness of our proposed module on the Fall Detection dataset.

	GMFS	CFE	mAP	mAR	AP50	AP75	APS	APM	APL	oLRP	
FGD			0.715	0.742	0.980	0.820	0.449	0.656	0.742	0.331	
1	√		0.717	0.754	0.981	0.813	0.439	0.661	0.743	0.328	
2		√	0.729	0.765	0.984	0.835	0.487	0.668	0.758	0.315	
3	√	√	0.731	0.768	0.981	0.837	0.461	0.670	0.760	0.311	
Notes.

GMFS Foreground separation based on Gaussian mask

CFE Channel feature extraction with probability distribution

Table 5 Ablation study on the effectiveness of our proposed module on the VOC2007 dataset.

	GMFS	CFE	mAP	
FGD			0.3365	
1	√		0.3470	
2		√	0.3850	
3	√	√	0.3904	
Notes.

GMFS Foreground separation based on Gaussian mask

CFE Channel feature extraction with probability distribution

Figure 8 Changes of mAP in terms of the number of iterations obtained by each compared algorithm on the (A) Fall Detection dataset and (B) VOC2007 dataset.

From Table 4, it can be observed that integrating each module individually enhances the performance of the method. Specifically, when the GMFS module is included in FGD, the mAP and mAR of the model increase by 0.2% and 1.2%, respectively, and oLRP decreases by 0.003; when the CFE module is integrated, the improvements are even more pronounced, with mAP and mAR increasing by 1.4% and 2.3%, respectively, and oLRP decreasing by 0.016. On the contrary, almost every result on each performance metric for model 1 or model 2 is worse than that of model 3 (i.e., FSD), verifying the effectiveness of the incorporation of both the GMFS and CWD modules. This similar phenomenon can also be observed in Table 5, demonstrating the validity of each module again.

The changes of mAP by each compared algorithm on two tested datasets indicate that both the two modules can improve the performance of FGD through the whole training process, especially the CFE module. Additionally, comparing the two lines of “FGD+CFE” and “FGD+GMFS+CFE” in Fig. 8B shows that the GMFS module can further enhance the stability of the changes of mAP.

Visual analysis

The feature maps enhanced with attention by all the compared KD methods are presented in Fig. 9, to provide a more intuitive understanding of the effectiveness of our proposed method. These images are sourced from the UR Fall Detection dataset. The maps obtained by the AT method may incorrectly extract non-target objects. The FT method is prone to background interference, which can degrade the visualization effects. The remaining compared KD methods, including FGD, MGD, CWD and BKCD, perform well from a global perspective, but compared to FSD, they show less distinct foreground features. Take the results obtained by CWD as examples. In the feature map produced by CWD for the first image, two holes appear in the middle of the human body, which are absent in FSD’s feature map. For the second image, the features extracted by FSD from the person’s feet are noticeably brighter compared to those extracted by CWD. In summary, it is evident that the FSD can better highlight target regions, thereby extracting target features more effectively and enhancing model performance.

Figure 9 Comparison of feature maps enhanced with attention of different KD methods.

Raw input image source: UR Fall Detection dataset.

Conclusions

This paper introduces an effective KD method tailored for the domain of object detection. The FSD method incorporates a foreground separation module based on Gaussian mask, to decouple foreground and background features. This approach not only filters out detrimental elements but also sharpens the model’s focus on foreground features. Based on the foreground separation result, the channel feature extraction using a probability distribution transforms spatial features into channel features, which are then fused with spatial features to improve the student model’s ability to capture details and reduce biases in foreground attention between teacher and student models. In addition, the attention mechanism adopted in this paper can further help the FSD method select critical regions and channels. The experimental results of various KD methods with the YOLOX detector and the ablation experimental results demonstrate the superiority of the FSD method and each module in it. In addition, since the FSD is a distillation method implemented on feature layers, it is applicable to both single-stage and two-stage object detectors. Future work will focus on optimizing the efficient utilization of background information within the FSD framework, as well as modifying the proposed method to enable its application to other tasks, such as semantic segmentation.

Additional Information and Declarations

Competing Interests

Author Contributions

Data Availability

The authors declare there are no competing interests.

Chao Li conceived and designed the experiments, analyzed the data, prepared figures and/or tables, authored or reviewed drafts of the article, and approved the final draft.

Rugui Liu conceived and designed the experiments, performed the experiments, performed the computation work, authored or reviewed drafts of the article, and approved the final draft.

Zhe Quan performed the experiments, performed the computation work, prepared figures and/or tables, authored or reviewed drafts of the article, and approved the final draft.

Pengpeng Hu performed the computation work, authored or reviewed drafts of the article, and approved the final draft.

Jun Sun analyzed the data, authored or reviewed drafts of the article, and approved the final draft.

The following information was supplied regarding data availability:

The raw data is available at Zenodo: Quan, Z. (2024). VOC and Fall [Data set]. Zenodo. https://doi.org/10.5281/zenodo.12752189.

The code is available at Zenodo: Quan, Z. (2024). SFC. Zenodo. https://doi.org/10.5281/zenodo.13829676.

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
