# Peer review of "Foreground separation knowledge distillation for object detection"

_PeerJ Computer Science, doi:10.7717/peerj-cs.2485_

## Round 0.1 · original submission · Major Revisions

Please revise the paper according to the reviewer's comments.

Reviewer 1 ·

Basic reporting

In this paper, the authors present an effective Knowledge Distillation (KD) method specifically designed for object detection. The proposed Foreground Separation Distillation (FSD) method includes a foreground separation module utilizing a Gaussian mask to decouple foreground and background features. This approach effectively filters out detrimental elements, enhancing the model's focus on foreground features. Additionally, the authors have improved the method by incorporating an attention mechanism, which selects the critical regions and channels, further refining the model's performance.

- The paper’s organization is perfect.
- The quality of the writing is excellent.

Experimental design

- The presentation methodology (i.e., methods, experiments, and analysis) is clear and understandable.
- The suggested approach's performances are well assessed using several experiments and comparisons as well as an ablation analysis.

Validity of the findings

- The obtained results are acceptable, competitive, and surpass all compared works.

Additional comments

- The references are recent and of good quality.

However, before accepting the manuscript, I would like that authors consider the following minor corrections:
- Check equation 4. Something is wrong.
- Line 205: Correct the following sentence « …so that so that we can use probability distance metrics…..” ----> « …so that we can use probability distance metrics…..” ---->
- Line 205: Correct the following sentence « … and the second term in the right part represents the represents the distillation loss between…..” ----> « … and the second term in the right part represents the distillation loss between…..” ---->

Cite this review as

Reviewer 2 ·

Basic reporting

All comments have been added in detail to the last section.

Experimental design

All comments have been added in detail to the last section.

Validity of the findings

All comments have been added in detail to the last section.

Additional comments

Review Report for PeerJ Computer Science
(Foreground separation knowledge distillation for object detection)

1. Within the scope of the study, a new knowledge distillation method is proposed for model compression in order to eliminate an important deficiency in the literature.

2. In the introduction section, the importance of the subject and the purpose of the study are clearly mentioned.

3. In the Related works section, although the basic literature on Focal and Global Knowledge Distillation is mentioned, in order to emphasize the difference of this study from the literature and its main contributions to the literature more clearly, it is suggested to add a table consisting of sections such as advantages, disadvantages, and results for the studies in the literature.

4. When the framework proposed in Figure-1 is examined in detail, it is observed that it has a certain level of originality. However, it is stated that the YOLOX model is used as the basis for detection. Although there are many different object detection models in the literature, please explain in more detail why this model is used as the basis and/or why different experiments are/are not performed.

5. It is stated that VOC2007 is used as the dataset in the use of the proposed method. When the literature is examined, the use of this dataset is deemed appropriate, but please detail in which areas studies can be conducted by referencing this method, especially in terms of future works.

6. Sharing the codes on the github platform regarding the study is positive and will probably further increase the usability of the method after publication.

7. Metrics such as AP and AR, which are preferred in terms of analysis of the results, are important evaluation metrics. However, there are also different metrics such as Optimal Localization Recall Precision (oLRP) that can be used for the analysis of object detection results in the literature. Please comment on your study from this perspective.

As a result, although the study has the potential to make a significant contribution to the literature, it is definitely recommended to pay attention to the above-mentioned sections.

Cite this review as

Reviewer 3 ·

Basic reporting

1. What are the main innovations in this paper? What is the role of knowledge distillation? The motivation, innovation and purpose of this paper should be clearly stated in the manuscript.
2. The introduction of the current research progress is insufficient, so it is recommended to introduce and analyze the latest research progress.

Experimental design

1. The results of the experiment are confused, so it is suggested that the analysis of the knowledge distillation part be more detailed.
2. It is suggested to increase the comparison with the SOTA methods to verify the effectiveness of the proposed method.

Validity of the findings

Statistical analysis of experimental results is lacking.

Cite this review as

---

## Round 0.2 · accepted · Accept

According to the comments of reviewers, after comprehensive consideration, it is decided to accept it.

Reviewer 1 ·

Basic reporting

The manuscript can be accepted in the current form.

Experimental design

/

Validity of the findings

/

Additional comments

/

Cite this review as

Reviewer 2 ·

Basic reporting

All comments have been added in detail to the last section.

Experimental design

All comments have been added in detail to the last section.

Validity of the findings

All comments have been added in detail to the last section.

Additional comments

Review Report for PeerJ Computer Science
(Foreground separation knowledge distillation for object detection)

Thanks for the revision. Since the responses to the reviewer comments and the changes in the paper were examined in detail and were observed to be sufficient, I recommend that the paper be accepted. I wish the authors success in their future projects. Best regards.

Cite this review as

Reviewer 3 ·

Basic reporting

no comment

Experimental design

no comment

Validity of the findings

no comment

Additional comments

no comment

Cite this review as